# Identification of *Haemaphysalis longicornis* Genes Differentially Expressed in Response to *Babesia microti* Infection

**DOI:** 10.3390/pathogens9050378

**Published:** 2020-05-14

**Authors:** Weiqing Zheng, Rika Umemiya-Shirafuji, Shengen Chen, Kiyoshi Okado, Paul Franck Adjou Moumouni, Hiroshi Suzuki, Shu Yang, Mingming Liu, Xuenan Xuan

**Affiliations:** 1The Collaboration Unit for Field Epidemiology of State Key Laboratory for Infectious Disease Prevention and Control, Jiangxi Provincial key Laboratory of Animal-Origin and Vector-Borne Diseases, Nanchang Center for Disease Control and Prevention, Honggutan New District, Nanchang 330038, China; zhengweiqing2001@gmail.com (W.Z.); chenshengen2020@outlook.com (S.C.); 2National Research Center for Protozoan Diseases, Obihiro University of Agriculture and Veterinary Medicine, Inada-Cho, Obihiro, Hokkaido 080-8555, Japan; okado@obihiro.ac.jp (K.O.); chakirou82@yahoo.fr (P.F.A.M.); hisuzuki@obihiro.ac.jp (H.S.); lmm_2010@hotmail.com (M.L.)

**Keywords:** *Haemaphysalis longicornis*, *Babesia microti*, vector-pathogen interactions, RNA interference, vaccine

## Abstract

*Haemaphysalis longicornis* is a tick and a vector of various pathogens, including the human pathogenetic *Babesia microti*. The objective of this study was to identify female *H. longicornis* genes differentially expressed in response to infection with *B. microti* Gray strain by using a suppression subtractive hybridization (SSH) procedure. A total of 302 randomly selected clones were sequenced and analyzed in the forward subtracted SSH cDNA library related to *Babesia* infection, and 110 clones in the reverse cDNA library. Gene ontology assignments and sequence analyses of tick sequences in the forward cDNA library showed that 14 genes were related to response to stimulus or/and immune system process, and 7 genes had the higher number of standardized sequences per kilobase (SPK). Subsequent real-time PCR detection showed that eight genes including those encoding for Obg-like ATPase 1 (ola1), Calreticulin (crt), vitellogenin 1 (Vg1) and Vg2 were up-regulated in fed ticks. Compared to uninfected ticks, infected ticks had six up-regulated genes, including *ola1*, *crt* and *Vg2*. Functional analysis of up-regulated genes in fed or *Babesia*-infected ticks by RNA interference showed that knockdown of *crt* and *Vg2* in infected ticks and knockdown of *ola1* in uninfected ticks accelerated engorgement. In contrast, *Vg1* knockdown in infected ticks had delayed engorgement. Knockdown of *crt* and *Vg1* in infected ticks decreased engorged female weight. *Vg2* knockdown reduced *B. microti* infection levels by 51% when compared with controls. The results reported here increase our understanding of roles of *H. longicornis* genes in blood feeding and *B. microti* infection.

## 1. Introduction

*Haemaphysalis longicornis* Neumann, 1901, is a three-host tick that occupies a broad ecological niche [1], resulting from its toleration of a wide range of environmental temperatures, its feeding on a wide range of mammalian and avian hosts, and its ability to survive various environmental conditions [2]. The tick was found in China, Japan, Korea and Russia and then invaded southern Pacific region countries, such as New Zealand and Australia, and now have established populations in the United States [3]. *H. longicornis* can reproduce through parthenogenesis, which facilitates its rapid emergence in nature, especially where there is a challenge for the ticks reproducing sexually to seek a sexual partner for mating [4]. The tick is an ectoparasite of importance in public health and domestic animal production, and it vectors a variety of pathogens such as bacteria, viruses and protozoa [5,6,7], which are transstadially and/or transovarially transmitted to other developmental stages [8,9].

Chemical acaricides have been the cornerstone of tick control for many years, but the application of chemicals on a vast and increasing scale has led to several serious adverse effects, including acaricide-resistance development [10,11]. It is reported that ticks develop resistance to several types of acaricides including organochlorines, organophosphates, carbamates, formamidines, pyrethroids, and macrocyclic lactones in many places [12]. To make things worse, ticks in some places have cross-drug resistance and a typical pattern of cross resistance has been documented among carbamates (such as carbaryl) and organophosphates (such as diazinon). In addition, tick resistance to more than one drug has also been shown in Mexico and Brazil [12,13]. To reduce their negative impact, a vaccine application may be an alternative means of tick control. Furthermore, a vaccine targeting both tick fitness and pathogen competence is an economical choice for the identification of tick molecules with a dual effect, namely simultaneous control of tick infestation and pathogen infection [14]. Suppression subtractive hybridization (SSH) is a technique for high-throughput anti-tick or/and pathogen vaccine screening and is considered to be an important procedure to find various vaccine candidates [15].

Until now, more than 100 *Babesia* species have been chosen to focus on due to the greatest emerging threats to animal health. High mortality and abortions caused by *Babesia* infection pose a great impact on livestock breeding around the globe [16,17,18]. In the case of human babesiosis, hundreds of cases have been reported in the world since the first case of human babesiosis was described in the United States in 1968. In the United States, the overwhelming majority of cases were caused by *Babesia microti*, and the rest by *B. duncani* and *B, divergens*-like organisms have also seen documented increases in recent years. In Europe, most reported cases have been caused by *B. divergens*, especially in splenectomized individuals, while a few have been attributed to *Babesia venatorum* (formerly called EU1) [19]. Sporadic cases of babesiosis incurred by *B. microti* and unidentified *Babesia* species have also been reported in Africa (Egypt), Australia, and South America [20]. Human babesiosis has been reported in Asian countries including Japan, Korea, India, and China. The *B. microti* Kobe Strain was responsible for human babesiosis in a Japanese patient who received a blood transfusion before the onset of babesiosis. Thereafter, the agent was also detected in blood donors who donated its blood to units where the patient was infected [21,22]. The emergence of human babesiosis in Korea has primarily been attributed to the contraction of a novel type of *Babesia* sp. (KO1) similar to ovine *Babesia* [23]. In India, a human babesiosis case is reported in a fifty-one-year-old male patient with a complaint of fever, anorexia, and vomiting [24]. In China, *B. microti* appears to be the dominant pathogen causing human babesiosis and it has caused babesiosis in more than 100 patients from Zhejiang, Yunnan, and Guangxi provinces over the past twenty years [25].

The objective of this study was to identify *H. longicornis* genes differentially expressed in response to *B. microti* infection and to evaluate tick gene functions in tick infestation and pathogen infection. Recently, we established an *H. longicornis*-*Babesia microti* infection model [26]. In the present study, differentially expressed genes in *B. microti*-infected *H. longicornis* ticks were screened by using the SSH procedure to subtract expressed genes in uninfected ticks. The results of the SSH studies were verified by real-time PCR. Functional analyses were carried out on selected *H. longicornis* genes by RNA interference (RNAi) to examine the potential role of these genes in blood feeding and *B. microti* burdens.

## 2. Results

### 2.1. Annotation of Forward and Reverse Subtracted SSH cDNA Library

The forward and reverse subtracted SSH cDNA libraries were successfully established (Figure A1, Figure A2, Figure A3, Figure A4 and Figure A5). Three hundred and eighty eight clones were randomly selected for sequencing and analyzed in the forward SSH cDNA library—the data concerning sequenced clones are presented in a Appendix A. After eliminating poor quality sequences, 302 sequences (average length ± S.D., 417 ± 183 bp) were assembled into 254 unigenes (32 contigs and 222 singlets) (Sheet “Forward subtracted cDNA library” in Appendix A). On average, the number of sequences per unigene was 1.19, which suggests a high diversity in our dataset. Sequence grouping resulted in 78 (25.83%) expressed sequence tags (ESTs) with an unknown function or without any identity to sequence databases. Significant identity to genes with functional annotation was confirmed for 224 sequences. They included the sequences from ticks, *Babesia*, hosts, and bacteria. Among them, ticks had the highest number of sequences, accounting for 61.59% (Figure 1).

Among the 186 tick gene sequences from the forward subtracted SSH cDNA library, 158 sequences (84.94%) were gene ontology (GO)-annotated with 86.71% sequences related to molecular function (MF), and 29.11% to biological process (BP) (Figure 2a). The highly represented BP included response to stimulus (39.13% of sequences) and metabolic process (39.13% of sequences) (Figure 2b). In the case of MF sequences, the gene class with a higher representation was catalytic activity (43.80%), followed by binding (21.90%) and transporter activity (18.25%) (Figure 2c). Catalytic activities were highly grouped into oxidoreductase activity (23.33%) and catalytic activity acting on protein (26.67%) (Figure 2d). The highly represented binding types were heterocyclic compound binding (46.67%) and protein binding (40.00%) (Figure 2e). The transporter activity sequences dominated by the lipid transporter activity genes (60.00%) were found (Figure 2f).

In the reverse subtracted SSH cDNA libraries, 143 clones were randomly selected, sequenced, and analyzed (Appendix A). After eliminating clones with poor quality sequences, 110 sequences (average length ± S.D., 341 ± 158 bp) were obtained. Sequence grouping resulted in 48 ESTs (43.46%) classified as “Unknown” due to an absence of annotation. Significant identity to genes with functional annotation was confirmed for 62 sequences, and the sequences from ticks were abundant with 59 ESTs (53.64%) (Figure 3).

GO analysis contributed to the annotation of 89.83% less expressed tick gene sequences (n = 53) in *B. microti*-infected *H. longicornis* (Figure 4a). The level 2 GO terms for BP and MF are shown in Figure 4b,c. The highly represented BP GO terms were localization (29.03%), response to stimulus (22.58%), and metabolic process (16.13%) (Figure 4b). Regarding the MFs, binding (37.50%) and catalytic activity (32.50%) were found to be highly represented (Figure 4c).

### 2.2. Analysis of Contig Repeats and Characterization of Response to Stimulus- and Immune System Process-Related Genes in Forward Suppression Subtraction cDNA Library

Out of 254 unigenes from the forward subtracted SSH cDNA library, 186 tick sequences were assembled into 150 unigenes (Sheet “Forward subtracted cDNA library” in Appendix A). Seven tick unigenes with a higher number of standardized sequences per kilobase (SPK) included *neutrophil elastase inhibitor* (*nei*), *cytochrome c oxidase VII*, *cytochrome b5, acyl-CoA-binding protein*, *Cathepsin L-like cysteine protease* (*clcp*), and *Cathepsin B-like cysteine protease* (*cbcp1*), and *nei* had the largest standardized SPK (Table 1).

After GO BP annotation of the tick genes highly expressed in response to *B. microti* infection, 14 genes were related to response to stimulus or/and immune system process. Two genes, namely *cbcp1* and *Ubiquitin-like protein FUBI* (*fubi*), play roles in both response to stimulus and immune system process. Nine genes including *calreticulin* (*crt*), *chemosensory protein CSP5*, *heat shock 70 kDa protein 5(hsp5)*, *nei*, *Obg-like ATPase 1(ola1)*, and *Vgs* (four vitellogenin analogues) were related to response to stimulus. Three genes, namely *leukocyte receptor cluster member 8 (lr8)*, *IgE-dependent histamine release factor (ihr)*, and *YTH domain-containing family protein 1(yth)* are responsible for immune system process (Table 2).

### 2.3. Expression Levels of the Less and Highly Represented Genes in Infected Ticks

Thirteen tick genes among the more expressed genes with putative functions in tick–pathogen interactions were selected on the basis of high standardized SPK value and association with immune system process and/or response to stimulus. The selection genes were then used for real-time PCR using total RNA from unfed, uninfected engorged, and *B. microti*-infected engorged ticks (Figure 5). These genes included *cbcp1*, *crt*, *fubi*, *hsp5*, *ihr*, *lr8*, *nei*, *ola1*, *Vg1*, *Vg2*, *VgDv*, *VgHf,* and *yth*. Of the genes analyzed, all except *lr8* and *yth* were confirmed as differentially expressed in fed ticks in comparison with unfed ticks. Genes encoding for *cbcp1*, *crt*, *hsp5*, *nei*, *ola1*, *VgHf*, *Vg1,* and *Vg2* were over-expressed in fed ticks while *fubi*, *ihr*, and *VgDv* mRNA levels were down-regulated in fed ticks. Compared to uninfected ticks, infected ticks had six upregulated genes, including *crt*, *hsp5*, *fubi*, *nei*, *ola1,* and *Vg2*, and three less-represented genes, namely *cbcp1*, *VgDv,* and *yth* (Figure 5).

### 2.4. RNA Interference Analysis of Upregulated Genes

The four genes confirmed to be highly expressed in infected or fed ticks (*ola1*, *crt*, *Vg1*, and *Vg2*) were selected for functional studies using dsRNA-mediated RNAi in *H. longicornis*. Under the conditions undertaken in this study, gene knockdown after dsRNA-mediated RNAi was demonstrated for all genes in *B. microti*-infected and uninfected *H. longicornis*, and silencing efficiencies of genes ranged from 65.83% to 96.48% in *B. microti*-infected ticks, and from 93.91% to 98.82% in uninfected ticks (Table 3).

The knockdown of *crt* and *Vg2* in infected ticks and *ola1* in uninfected ticks accelerated the engorgement of the ticks. However, *Vg1* silencing in infected ticks delayed the engorgement. *Crt* and *Vg1* knockdown in the infected ticks decreased the engorged female weight (EFW) of the ticks (Table 3). *Vg2* knockdown reduced the *B. microti* infection level by 51%. For other genes such as *ola1*, *crt* and *Vg1*, gene knockdown did not alter *B. microti* infection levels compared with controls (Table 4).

## 3. Discussion

Several studies on pathogen–tick interactions have showed that pathogens could modulate tick molecules for infection [27]. A pathogen might induce cytoskeletal rearrangement to maintain infection. For example, the alteration of the ratio between monomeric globular *G actin* and filamentous F actin of *Ixodes scapularis* has been shown to facilitate *Anaplasma phagocytophilum* infection. Likewise, *A. phagocytophilum* infection has been linked to the up-regulation of *spectrin alpha chain* or *Alpha-fodrin* of *I. scapularis* [28,29,30]. In the present study, we also found that *spectrin alpha chain* transcripts were over-expressed *in B. microti*-infected *H. longicornis* ticks. Additionally, previous studies showed that *A. phagocytophilum* may benefit from the tick cells’ ability to limit pathogen infection through phosphoenolpyruvate carboxykinase (PEPCK) and voltage-dependent anion channel (VDAC) inhibition [28,30,31,32,33]. Other tick molecules facilitating pathogen acquisition include Kunitz-type protease inhibitors, Bm86, subolesin, crt, and serum amyloid A [34]. However, when ticks are infected by a pathogen, they activate an immune or stress response to combat against pathogen infection including heat shock protein (HSP) [34,35,36]. HSP response might help to increase tick survival by protecting from stress and preventing desiccation at high temperatures. In agreement with those reports, our previous data showed that VDAC was less expressed in *B. microti*-infected engorged female *H. longicornis* ticks [26] and in the present study, HSP was over-represented in the transcripts of *B. microti*-infected engorged female *H. longicornis* ticks. In addition, our results indicate that ola1, crt, and vitellogenin family proteins such as Vg1, Vg2 may have important roles in blood feeding and *Babesia* infection in *H. longicornis* female ticks.

The GTPase superfamily proteins interact with rRNA and/or ribosomes to bind and hydrolyze GTP through the P-loop NTPase fold. They are involved in many cellular processes including protein synthesis, vesicular traffic, intracellular transport, cell signaling, and differentiation [37,38,39]. *Ola1* shares some of the most important features including the G domain of the *GTPases* [40]. However, unlike other GTPases, the *ola1* protein preferentially binds to and hydrolyzes ATP rather than GTP [41]. Ola1 acts as a molecular chaperone to regulate cellular stress responses, including oxidative stress and heat shock [42,43,44]. In addition, ola1 regulates cell adhesion and migration, and cellular apoptosis through the polymerization and S-glutathionylation of *actin*, and mediation of anti-apoptotic proteins such as cellular inhibitor of apoptosis protein 1 (cIAP1), cIAP2, and B cell lymphoma extra-large (Bcl-xL), respectively [45,46]. The knockdown of *ola1* mRNA by RNAi inhibits the growth of procyclic forms of the protozoan *Trypanosoma cruzi* [40,43]. In our study, *H. longicornis* ticks fed on *B. microti*-infected hamsters, confronted two adverse conditions, namely, oxidative stress resulting from blood feeding, and *Babesia* infection. The two conditions might have upregulated *ola1* expression in infected fed ticks, which likely explains why the *ola1* transcript was present in the forward suppression subtraction cDNA library.

Crts are secreted in tick saliva and the saliva of other blood-sucking arthropods. They are suggested to play a critical role in blood feeding by preventing blood clotting through binding to Ca^2+^ and clotting factors [47,48,49,50]. In our study, *crt* mRNA was upregulated in engorged/infected ticks, and *crt* dsRNA injection to replete ticks infected with *Babesia* caused a reduction in EFW and shortened BFP. On the other hand, *crt*s have also been reported to have a role in response to infection. Crts from some parasites are known to bind to the first component of the classical complement system (C1q) to maintain an immunosuppressive and anticomplement state of the infested animals for the parasite’s survival [51]. It was demonstrated that vaccination of sheep with *Haemaphysalis qinghaiensis* recombinant crt (rHqcrt) conferred protective immunity against *H. qinghaiensis* ticks [52]. In contrast, despite high amino acid identity to homologs in other organisms, *Amblyomma americanum* tick calreticulin (Aamcrt) is apparently functionally different. Aamcrt bind to C1q, but does not inhibit activation of this pathway. Furthermore, recombinant Aamcrt does not bind to factor *Xa*, an important protease in the blood-clotting system [53]. In addition, feeding with artificial blood supplemented with anti-r*crt* serum does not cause significant reductions in tick and egg weight in *R. microplus*. Crt protein has also been identified as being involved in *B. bigemina* infection in *R. annulatus*, and *crt* knockdown had a significant effect on the pathogen infection in *R. microplus*, but not in *R. annulatus* ticks. Furthermore, *B. bigemina* infection levels did not show any statistically significant decrease when ticks were fed with anti-rcrt antibodies [54]. Herein, *crt* mRNA was upregulated in *B. microti*-infected ticks, but *crt* dsRNA injection did not change *Babesia* burdens. Taken together, our data and previous studies suggest that *crt*s from different tick species may have complex machineries in association with blood feeding and *Babesia* infection in hosts.

Vgs, the precursor proteins of egg yolk vitellin (Vn), are largely present in females of all animals with yolky eggs and thus have been called female-specific proteins. Vg might participate in defense reactions and possesses antibacterial activities, which is likely explained by the involvement of the molecule in damaging bacterial cell walls, reducing nuclear translocation of the forkhead transcription factor DAF-16, and increasing anti-oxidative capacity in a steroidal-signaling pathway-dependent manner [44,55,56]. Furthermore, these reproductive proteins act as a pattern recognition molecule capable of binding to fungi and enhancing macrophage phagocytosis, high concentrations of Vgs in host hemolymph may help the host neutralize an invading virus, and nematode and protozoan infections are likely to transform the patterns of *Vg* mRNA transcription and protein translation in hosts [57,58,59,60,61]. Three homologs of *Vgs* namely *Vg1*, *Vg2*, and *Vg3* have previously been described in *H. longicornis*. When *H. longicornis* ticks suck blood, *Vg1* is expressed in the midgut and *Vg2* in the fat body and ovary [62]. In *Babesia*-infected ticks, Vg is intensively produced in basophilic cells of the midgut which *Babesia* protozoans are known to infect [63]. In the present study, Vg1, Vg2, and two other homologs of *Vgs* were present in the transcripts of the infected *H. longicornis* ticks after subtraction of those of the uninfected ticks. However, only *Vg2* was highly expressed in *B. microti*-infected ticks, and *Vg2* knockdown shortened BFP and reduced *B. microti* infection levels. Further clarification of how Vg1 and Vg2 affect *Babesia* invasion and binding, and penetration into host cells and migration from the midgut to other organs like the ovary and salivary gland is required. In future studies, we will characterize the expression profiles of *Vg1* and *Vg2* in different organs of the infected ticks, such as *Vg1* in the midgut and *Vg2* in the fat body and ovary.

## 4. Materials and Methods

### 4.1. Unfed, Uninfected, and B. microti-Infected Ticks

In our study, the uninfected/fed ticks designate female *H. longicornis* ticks infesting and engorging on the uninfected hamsters, and the *B. microti*-infected ticks denote female *H. longicornis* ticks infesting and engorging on the *B. microti*-infected hamsters. *B. microti*-infected and -uninfected ticks were produced to construct SSH libraries. Unfed or uninfected ticks were used for comparison to uninfected or infected ticks respectively for analyses of mRNA expression levels. Parthenogenetic *H. longicornis* (Okayama strain) ticks have been maintained by feeding on the ears of Japanese white rabbits (Japan SLC, Shizuoka, Japan) for generations at the National Research Center for Protozoan Diseases (NRCPD), Obihiro University of Agriculture and Veterinary Medicine, Obihiro, Japan [64]. To obtain *B. microti*-infected ticks, 8-week-old female Syrian hamsters (Japan SLC, Shizuoka, Japan) were administrated orally with a daily dosage of 2 mg of cortisone for three days, and thereafter inoculated intraperitoneally with *B. microti* (Gray strain) parasites using the methods mentioned in The Global Bioresource Center (ATCC^®^ 30221™). When approximately 5% parasitemia was reached, 12 female ticks were put into a chamber attached to the shaved back of a Syrian hamster, and the practice was repeated three times. Conventional PCR using KOD-Plus-Neo DNA polymerase (Toyobo, Osaka, Japan) and *B. microti β-tubulin*-specific primers and *H. longicornis actin*-specific primers (control gene) was performed on nymphal samples to detect *B. microti* DNA [26]. Simultaneously, three naïve hamsters were used to obtain uninfected female ticks. Unfed female ticks were collected from the same batch as uninfected and *B. microti*-infected ticks, and kept at 25 °C under air at 85–90% RH pending the test. The rabbits and hamsters were reared in a temperature- and humidity-regulated room under controlled lighting, water, and regular commercial chow. All procedures were carried out according to ethical guidelines for the use of animal and pathogen samples permitted by Obihiro University of Agriculture and Veterinary Medicine (Animal exp.: 19-74 for rabbits and 19-77 for hamsters; DNA experiment approval number: 1704-4; pathogen experiment approval number: 201709-5).

### 4.2. SSH cDNA Library Construction

*B. microti*-infected and -uninfected ticks were used to construct the forward and reverse subtracted SSH libraries. Engorged female ticks were thrice rinsed individually in distilled water, once in water, once in 70% (*v*/*v*) ethanol, and once more in water. Engorged ticks of a similar size were screened by body weight and each was dissected individually in cold nuclease-free PBS buffer. Host blood in tick body was removed as much as possible by washing the midgut tissues several times with cold nuclease-free PBS buffer. After several washes, whole internal organs were ground in a 1.5 mL sterile plastic tube with 200 µL of RA1 cell lysis buffer (Macherey-Nagel GmbH & Co KG, Duren, Germany) and 2 µL of β-mercaptoethanol. Homogenates from 15 ticks of the *B. microti*-infected or uninfected group were pooled in a 15 mL sterile tube, and 100 µL of homogenate was collected into a 1.5 mL tube for DNA isolation by using NucleoSpin^®^ Tissue Kit (Macherey-Nagel GmbH & Co KG, Duren, Germany). A mixture containing 3 mL of RA1 cell lysis buffer plus 30 µL of β-mercaptoethanol was added to the rest of the homogenate and the mixture was used to isolate total RNA with NucleoSpin^®^ RNA Midi Kit (Macherey-Nagel GmbH & Co KG, Duren, Germany). Total RNA and genomic DNA were isolated according to the manufacturer’s protocol. The DNA sample was subjected to conventional PCR for detection of *B. microti* burdens in the sampled ticks using *B. microti β-tubulin*-specific primers (Table A1) [65]. RNA was quantified with Thermo Scientific™ NanoDrop 2000 (Thermo Fisher Scientific Inc., Waltham, MA, USA) and its quality was examined by gel electrophoreses to confirm the integrity of RNA preparations. Two pools corresponding to the infected and uninfected tick populations were made. Poly A^+^ RNA was separated from total RNA using NucleoTrap^®^ mRNA Midi Kit (Macherey-Nagel GmbH & Co KG, Duren, Germany) and concentrated with NucleoSpin^®^ RNA Clean-up XS (Macherey-Nagel GmbH & Co KG, Duren, Germany). The cDNA was synthesized and the SSH library was constructed using the PCR-Select™ cDNA subtraction Kit (Clontech, Mountain View, CA, USA). Briefly, double-stranded cDNA from both groups (infected and uninfected ticks) was digested with RsaI. Two µL of the digested cDNA was ligated with 2 µL of 10 µM Adapter 1 in a mixture containing 3 µL of sterile H_2_O, 2 µL of 5X Ligation Buffer and 1 µL of T4 DNA Ligase (400 units/µL) at 16 °C overnight, and 2 µL of the other digested cDNA was ligated with 2 µL of 10 µM Adapter 2R in the same manner. Ligation efficiency analysis was performed by PCR with PCR primer 1 and primers (Table A1) targeting *porin* which is expressed in infected and uninfected engorged female ticks [14]. The remainder of the cDNA was saved to be used as a driver in hybridization. The forward subtracted library was made by hybridizing adapter-ligated cDNA from *B. microti*-infected ticks as the tester in the presence of surplus uninfected tick cDNA as the driver. The reverse subtracted library was prepared by hybridizing adapter-ligated cDNA from uninfected ticks as the tester with excess *B. microti*-infected tick cDNA as the driver. Analysis of subtraction efficiency was carried out by PCR with a primer set showed in Table A1. Differentially expressed cDNAs were subject to primary and secondary PCR with Advantage PCR polymerase mix, and cloned using pGEM^®^-T Easy Vector Systems (Promega, Madison, WI, USA) for sequencing.

### 4.3. Gene Ontology Analysis

Nucleotide sequences of forward and reverse subtracted libraries clones were trimmed and utilized for annotation and gene ontology (GO) assignment. Plasmid DNAs were purified using the NucleoSpin^®^ Plasmid EasyPure Kit (Macherey-Nagel GmbH & Co KG, Duren, Germany). Three hundred and eighty eight clones from the forward subtracted library and 143 clones from the reverse subtracted library were randomly selected and sequenced with primer set SP6 and T7 on an ABI PRISM 3100 DNA sequencer (Applied Biosystems, Waltham, MA, USA). Raw sequence data were loaded to DNASTAR SeqMan and vector contamination filtering was done by scanning each fragment with the preassembly and assembly options program. The fragments without the pGEM^®^-T Easy vector were manually trimmed to remove adaptors at 5′ and 3′ ends. Trimmed nucleotide sequences with sizes greater than 80 bp were blasted against online sequence databases of National Center for Biotechnology Information (NCBI) [66], Kyoto Encyclopedia of Genes and Genomes (KEGG) [67], and Bioinformatics Resource for Invertebrate Vectors of Human Pathogens (VectorBase) [68]. An E value <10^−10^ was set as a condition for successful gene assignments. GO terms were assigned to differentially expressed genes with the online Software Tool MGI (http://www.informatics.jax.org/vocab/gene_ontology), Uniprot (https://www.uniprot.org/) (version 2020), and Quick GO (EMBL-EBI) (https://www.ebi.ac.uk/QuickGO/) (version 2020). For analysis, two categories of GO terms, namely Biological Process (BP) and Molecular Function (MF) were evaluated.

### 4.4. Sequence Analysis

For the unigene cluster formation, the nucleotide sequences from NCBI, KEGG, and VectorBase databases with hits to the nucleotide sequences in the forward suppression subtraction cDNA library, were compared with the nucleotide sequences in the cDNA library. For genes with more than one sequence, the longer sequence was considered. The sequences per kilobase (SPK) was calculated to normalize the sequence counts for each unigene according to gene length. SPK of a unigene was calculated as the sequence counts divided by the length of the longer sequence in kilobase. Standardized SPK for a unigene was computed as SPK of the unigene divided by total SPK (Sheet “Forward subtracted cDNA library” in Appendix A). After gene ontology assignments, genes related to a response to stimulus or immune system process in the forward suppression subtraction cDNA library were screened and characterized.

### 4.5. Real-Time PCR for Analysis of Differential Gene Expression

The expression levels of screened genes were analyzed in unfed, fed/uninfected, or *B. microti*-infected ticks. Three independent biological replicates were made for each group of tick samples. After two washes with double distilled water and one wash with 70% ethanol, 3 unfed female ticks and 3 engorged female ticks with host blood removed were homogenized in TRI reagent (Sigma-Aldrich, St. Louis, MO, USA). RNA extraction, cDNA synthesis, and real-time PCR were performed as described elsewhere [64]. *Glyceraldehyde-3-phosphate dehydrogenase* (*GAPDH*) was used as an internal control gene in real-time PCR analysis [26] and all primers are shown in Appendix B
Table A1. The PCR reaction system was set as follows: 5% of the volume of cDNA solution, 0.5 μM of the forward and reverse primers, 10 μL of THUNDERBIRD^®^SYBR^®^qPCR Mix (Toyobo, Osaka, Japan, and 0.04 μL of 50X ROX reference dye, and sterile water was added up to a final volume of 20 μL. The PCR conditions were as follows: 95 °C for 10 min, 40 cycles of denaturation at 95 °C for 15 s, and annealing/extension at the temperatures shown in Table A1 for 35 s. The mRNA levels were normalized separately against mRNA levels of internal control gene using the ∆CT (2^^(−(CT_target gene_−CT_internal control gene_))^) method.

### 4.6. RNA Interference for Functional Analysis of Tick Genes Differentially Expressed in Response to Babesia Infection

RNA interference (RNAi) was used to analyze the effect of target gene knockdown on blood feeding and *Babesia* infection. The double-strand RNA (dsRNA) of target genes was constructed with the T7 RiboMAX express RNAi system (Promega, Madison, WI, USA) and the primer sets shown in Table A1. Female ticks were used for RNAi experiments [69]. In the test group, unfed female ticks were injected with 1 µg of target gene dsRNA/tick in 0.5 µL of RNase-free water at the fourth coxae using a grass capillary equipped with IM-300 Microinjector (Narishige, Tokyo, Japan). The same amount of firefly *luciferase* dsRNA constructed from a vector DNA of pGEM-luc (Promega, Madison, WI, USA) was injected into unfed ticks as a control [70]. The dsRNA-injected ticks were allowed to rest 1 day and were then put in chambers attached to the back of a hair-shaved hamster. Each hamster was challenged with 15 dsRNA-injected ticks for the control group or test group, and two hamsters were used in each group. RNA silencing efficiency was evaluated by comparison of the normalized mRNA levels of target genes against *GAPDH* between the test group and the control group. In order to examine knockdown efficiency after dsRNA injection, two 2-day-fed ticks were removed from each of the infested hamsters and the practice was repeated three times. Engorged ticks were dissected to collect whole internal organs. Host blood in the midguts was removed as much as possible by washing the internal organ tissues several times with cold PBS, and resultant organs were ground using a pestle. In order to test whether silencing of target genes affected *Babesia* infection levels, *B. microti* burdens assay was performed. This was done by *B. microti* 18S rRNA real-time PCR using genomic DNA from the whole internal organs of three engorged ticks, and duplicated thrice. The genetic amounts of *B. microti* 18S rRNA were normalized against that of *H. longicornis* ITS-2 (Table A1) using the ∆CT (2^^(−(CT_target gene_−CT_internal control gene_))^) method. *B. microti* infection levels were calculated using the relative amount of *Babesia* 18S rRNA against tick ITS-2 in infected ticks, standardized by those observed in uninfected ones. The feeding success of ticks was investigated by measuring the blood feeding period (BFP) and engorged female weight (EFW).

### 4.7. Statistical Analysis

In all experiments, the mean of relative mRNA levels of target genes was computed and data from unfed, uninfected, and infected ticks were compared using the Student’s *t*-test. The differences in the mean of *B. microti* burdens and EFW between target gene dsRNA-injected group and *luciferase* dsRNA-injected group were also analyzed with the Student’s *t*-test. The difference in BFP between the dsRNA-injected and control groups was analyzed using the Mann-Whitney test. A *p* value of < 0.05 was considered statistically significant.

## 5. Conclusions

In the present study, many genes from *H. longicornis* were involved in *B. microti* infection. These genes included *cbcp1*, *crt*, *fubi*, *hsp5*, *ihr*, *lr8*, *nei*, *ola1*, *Vg1*, *Vg2*, *VgDv*, *VgHf*, and *yth.* Among them, *ola1*, *crt*, *Vg1*, and *Vg2* could contribute to the interference with blood feeding and *Vg2* might assist *B. microti* infection in ticks.

## Figures and Tables

**Figure 1 pathogens-09-00378-f001:**
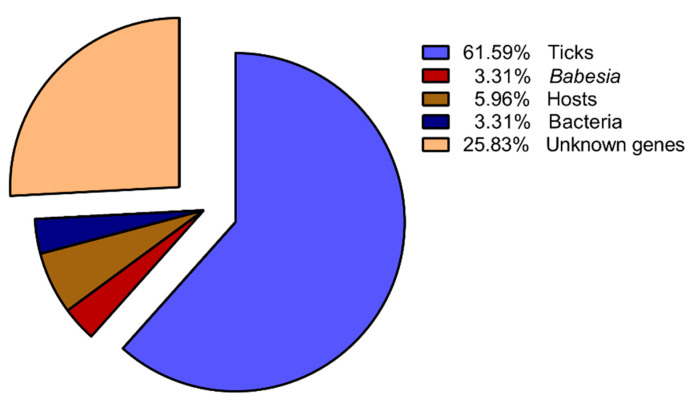
Representation of up-regulated tick sequences in *B. microti*-infected *H. longicornis* females.

**Figure 2 pathogens-09-00378-f002:**
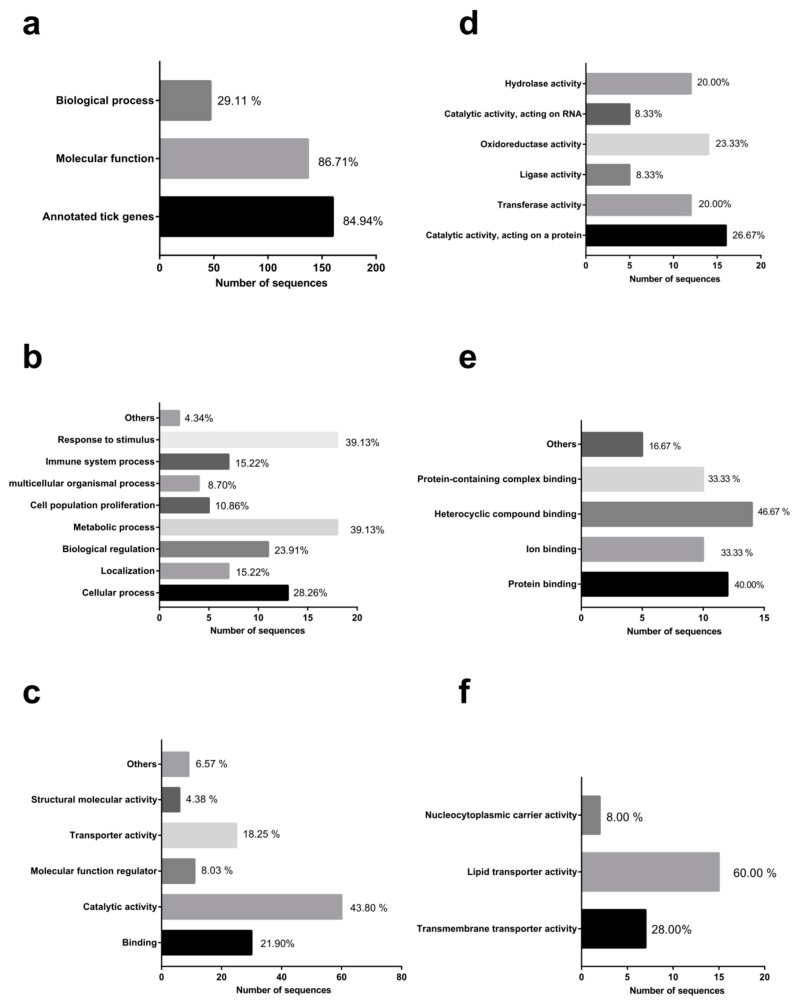
Functional grouping of tick genes highly expressed in *B. microti*-infected *H. longicornis* based on gene ontology (GO), molecular function (MF), and biological process (BP) assignments. (**a**) GO, MF, and BP assignments. (**b**) GO and BP assignments (level 2). (**c**) GO and MF assignments (level 2). (**d**) GO catalytic activity assignments. (**e**) GO binding assignments. (**f**) GO transporter activity assignments.

**Figure 3 pathogens-09-00378-f003:**
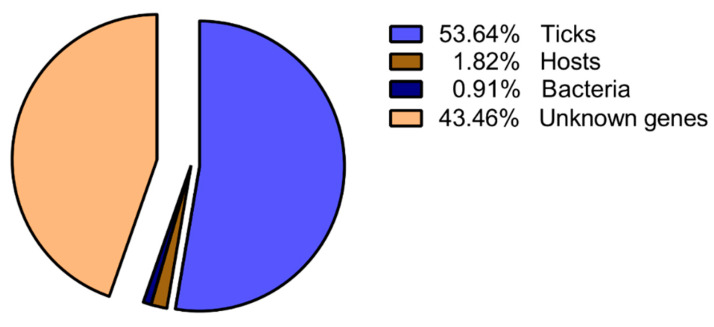
Representation of down-regulated tick sequences in *B. microti*-infected *H. longicornis* females.

**Figure 4 pathogens-09-00378-f004:**
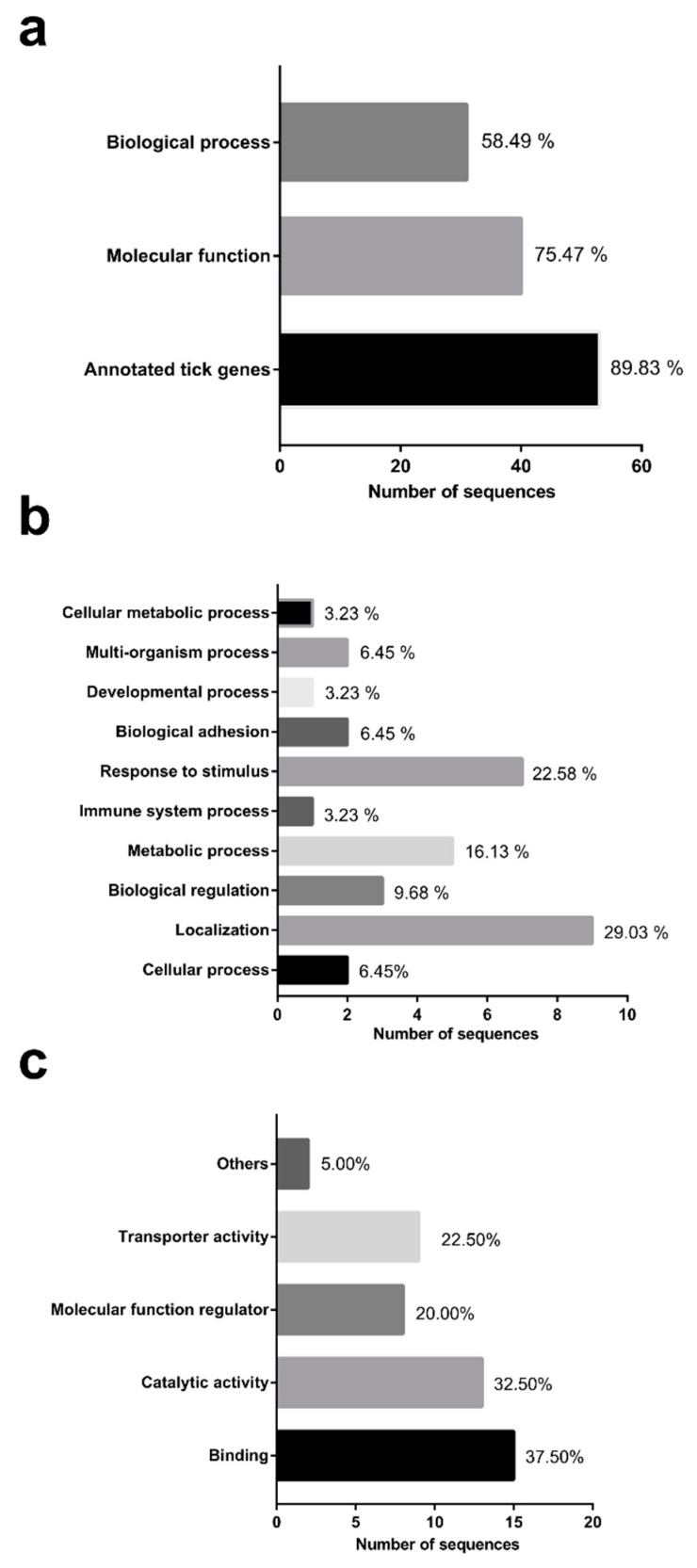
Functional grouping of tick genes less expressed in *B. microti*-infected *H. longicornis* based on GO, MF, and BP assignments. (**a**) GO, MF, and BP assignments. (**b**) GO and BP assignments (level 2). (**c**) GO and MF assignments (level 2).

**Figure 5 pathogens-09-00378-f005:**
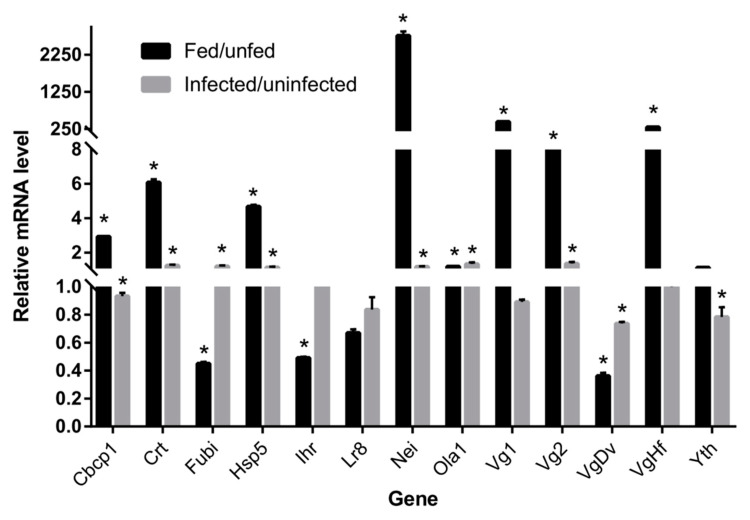
Differential gene expression in *Babesia microti*-infected and -uninfected *Haemaphysalis longicornis* ticks. The mRNA levels of selected differentially expressed genes were determined by real-time reverse transcription-PCR. The mRNA levels were normalized against tick *GAPDH* transcripts using the ddCT method (2^(−(CT_target_−CT*_GAPDH_*))). The infected/uninfected (fed/unfed) ratios represent the expression fold-change in the infected (fed) ticks calculated by dividing the mean of normalized mRNA levels in infected ticks (three ticks as a biological replicate, three replicates were done) by the mean of the normalized mRNA level in uninfected (unfed) control ticks (nine ticks in three independent biological replicates). In all tests, the mean of the duplicate values was used and data from *B. microti*-infected and uninfected ticks were compared using the Student’s *t*-test. * *p* < 0.05.

**Table 1 pathogens-09-00378-t001:** The unigene expression pattern in forward suppression subtraction cDNA library.

Unigene(s)	Sequences for the Unigene (FSN ^1^)	Standardized SPK ^2^
*Neutrophil elastase inhibitor (nei)*	5 sequences (46,60, 99, 203, 371)	0.0717
*Cytochrome c oxidase subunit VII*	1 sequence (87)	0.0217
*Cathepsin L-like cysteine protease (clcp)*	2 sequences (15, 31)	0.0206
*Acyl-CoA-binding protein*	1 sequence (287)	0.0177
*Putative secreted salivary peptide*	1 sequence (220)	0.0156
*Cytochrome b5*	2 sequences (368, 388)	0.0151
*Cathepsin B-like cysteine protease (cbcp1)*	2 sequences (105, 380)	0.0132

^1^ FSN, clone number in the forward suppression subtractive hybridization (SSH) cDNA library of Appendix A. The numbers in the bracket are clone-specific numbers in the forward SSH cDNA. ^2^ SPK, standardized sequences per kilobase.

**Table 2 pathogens-09-00378-t002:** Response to stimulus- and immune system process-related tick genes in forward suppression subtraction cDNA library.

Gene	Function
*Calreticulin* (*crt*)	Response to stimulus
*Cbcp1*	Response to stimulus/immune system process
*Chemosensory protein CSP5*	Response to stimulus
*Leukocyte receptor cluster member 8* (*lr8*)	Immune system process
*IgE-dependent histamine release factor* (*ihr*)	Immune system process
*Heat shock 70 kDa protein 5(hsp5)*	Response to stimulus
*Nei*	Response to stimulus
*Obg-like ATPase 1(ola1)*	Response to stimulus
*Ubiquitin-like protein FUBI* (*fubi*)	Response to stimulus/immune system process
*Vgs* ^1^	Response to stimulus
*YTH domain-containing family protein 1(yth)*	Immune system process

^1^
*Vgs include H. longicornis vitellogenin 1, H. longicornis vitellogenin 2, Haemaphysalis flava vitellogenin-like Vg(VgHf), and Dermacentor variabilis vitellogenin-like Vg (VgDv).*

**Table 3 pathogens-09-00378-t003:** Evaluation of engorged female weight and blood feeding period of *B. microti*-infected and -uninfected *H. longicornis* female ticks injected with specific dsRNA.

Group	Target Genes for RNAi	*n*	Gene Silencing (%) (Mean ± S.D.)	EFW (mg) (Mean ± S.D.)	BFP (d) (25% Percentile–75% Percentile)
Infected	*Crt*	16	84.97 ± 2.67	65.44 ± 22.32 ^a^	6.00–7.00 ^a^
	*Ola1*	16	96.48 ± 0.37	116.26 ± 12.58	6.75–9.00
	*Vg1*	14	65.83 ± 2.46	68.71 ± 20.77 ^a^	7.00–9.00 ^a^
	*Vg2*	16	82.04 ± 2.37	110.89 ± 26.03	6.00–6.00 ^a^
	Control	16	-	101.30 ± 7.84	7.00–7.00
Uninfected	*Crt*	12	93.91 ± 1.23	75.88 ± 5.60	5.75–7.00
	*Ola1*	12	97.44 ± 0.74	88.30 ± 6.44	5.00–5.75 ^a^
	*Vg1*	14	98.82 ± 0.13	92.06 ± 7.62	6.00–7.00
	Control	16	-	82.52 ± 13.78	6.25–7.75

Gene knockdown was analyzed by real-time PCR in six ticks per group by comparing mRNA levels between specific dsRNA-injected and control ticks. EFW, engorged female weight; BFP, blood feeding period; ^a^, statistical difference (*p* < 0.05) was analyzed in EFW by the Student′s *t* test and in BFP by the Mann-Whitney test when the dsRNA-injected group was compared to the control group.

**Table 4 pathogens-09-00378-t004:** *Babesia microti* infection levels after gene knockdown by RNAi in *H. longicornis* ticks.

Gene	*B. microti* Burdens (Mean ± S.D.)	Test/Control (Mean ± S.D.)
*Crt*	0.26 ± 0.06	0.95 ± 0.21
*Ola1*	0.34 ± 0.08	1.25 ± 0.29
*Vg 1*	0.26 ± 0.08	0.94 ± 0.22
*Vg 2*	0.14 ± 0.03	0.51 ± 0.10 ^a^
*Control*	0.27 ± 0.05	-

^a^, statistical difference (*p* < 0.05) when compared to the control group.

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
