# Peer review of "Identification of Haemaphysalis longicornis Genes Differentially Expressed in Response to Babesia microti Infection"

_pathogens, 2020, doi:10.3390/pathogens9050378_

Round 1

Reviewer 1 Report

The manuscript by Zheng et al reports the expression of genes potential involved in Babesia-tick interactions. The authors used RNAi to functionally characterized some of the genes identified. The manuscript involved a heavy experimental work. However, the description of the results can be improved. The authors should address some points before considering the manuscript for publication.

Introduction

Line 37: Introduction about Babesia is missing.

Lines 48-56: The connection between tick resistance to acaricides and mites resistance to acaricides in the text is not well-established. Please, reword.

Line 51: 'mites confer resistance ... ' or 'mites develop resistance ...' ?

Line 62: The effect of gene silencing (using dsRNA) on ticks cannot be extrapolated to vaccine efficacy. In the present study, the finding of 'vaccine candidates' is not addressed. Therefore, the aim of the study has to be reworded.

Line 64: Not clear what 'motivator' means in this context.

 Results

Line 72: If the original method of SSH by Diatchenko et al., (Proc Natl Acad Sci U S A. 1996 Jun 11; 93(12): 6025–6030. ) was used, then please, specify here that this library contained clones only present in Babesia-infected ticks. If this is not the case, specify the content of this library.

Line 73: Supplementary files have to be named as Figures S1, 2 / Table S1, 2 and so on. Not 'ESM.xlsx'. Check this in the whole manuscript.

Line 81: However, at the end of the document (line 415), there is another figure labeled as 'figure 1'. Please, check this. Same for figures 2-4.

Line 83: 'highly-expressed' or 'highly-represented transcripts in the library'. High expression makes reference to expression levels and this is not what is displayed in this figure. Same in the legend of Figure 3.

Lines 92-93: This sentence needs revision, the expressions 'were dominated/were found' are not grammatically correct in this context and makes the sentence difficult to understand.

Line 95: The letters of the panels (a, b, c…) should go top/left, not in the middle. Also, the size of the numbers/letters is not the same in all the panels. Same for figure 4.

Line 121: Spell the abbreviations (i.e. NEI) the first time they are used in the text. Name of genes go in Italics and low case.

Line 122: ribosomal protein subunit genes are a common contamination is this type of study because they are highly abundant in the cells. What the authors did to ensure that this bias would not affect their analysis?

Line 127: It is not clear why, If the authors will move forward with the genes that resulted from the GO annotation analysis, they introduced above the genes with highest 'sequences per kilobase (SPK)' (lines 120 - 126). If these genes with the highest 'sequences per kilobase (SPK)' were not going to be used in further analysis, then, better to move table 1 to supplementary materials.

Lines 127-131: Not clear the criteria used by the authors to select these genes.

Line 128: Change 'are' to 'were'

Line 130: How the authors confirmed that these four vitellogenin genes are homologues and not paralogues?

Line 140: name of genes in Italics and low case.

Line 143: Not clear why the authors switched here to test the expression of the genes in fed ticks. The focus of the paper is Babesia infection and not feeding. Also, gene expression results are not easily visualized as a table. Present these results in a Figure.

Line 157: In this section, it should be clearly stated what was the control used, 'firefly luciferase dsRNA-injected group' or 'non-injected group' or what ?

Discussion

Line 183: Please, add reference to the original manuscripts referring to the modification in the expression of cytoskeleton genes upon Anaplasma phagocytophilum infection (Front Biosci (Landmark Ed). 2017;22:1830-1844. & Infect. Immun. 2013. 81, 2415–2425).

Line 190: Here a research manuscript referring to the modification in the expression of HSPs genes upon Anaplasma phagocytophilum infection (https://link.springer.com/chapter/10.1007/978-3-319-73377-7_15)

Lines 234-253: Maybe a sentence can be added about how upregulation of Vg could be exploited by Babesia for transovarial transmission. Binding of Babesia to Vg can be the route this pathogen uses to enter tick eggs.

Methods

Line 255: Not clear in this section how the authors confirmed that the ticks acquired Babesia after feeding on Babesia-infected animals.

Line 266: There is information missing here. How the Babesia was maintained? What infectious dose of Babesia was used?

Line 267: How parasitemia was quantified?

Lines 268-269: Not clear here the proportion tick/hamster used for pathogen acquisition experiments? 12 ticks per hamster or 12 ticks for the 3 hamster? 4 ticks per hamster?

Line 271: Detail the conditions used for tick incubation.

Line 346: Are the authors referring to 'Three independent biological replicates' ?

Line 353: Change to 'gene'. Only one internal control gene was used (Line 350).

Lines 363-364: It is not clear what the authors did here. The expression 'target genes against gapdh between target gene ….' is not clear. Reword and explain better.

Conclusions

The conclusions should be further elaborated. Please, be more specific than 'The present study adds evidence to our knowledge… '. What about the conclusions concerning the genes involved in the tick response to Babesia infection?

Tables

Table 1: Not clear what the numbers inside brackets are. Also, what the abbreviation 'FSN' stand for?

Author Response

Dear peer reviewer,

    We express our gratitude to you for the critical review and giving constructive suggestions to improve our manuscript. We have heeded your suggestions and have rearranged our manuscript to further clarify and strengthen the findings. Our point-by-point response to your specific comments are provided in the attachment. By the way, we also have a revised version of our manuscript and submit our documents with words in red font to highlight the revisions. 

Sincerely yours,

Xuenan Xuan, PhD

National Research Center for Protozoan Diseases

Obihiro University of Agriculture and Veterinary Medicine

Inada‑Cho, Obihiro, Hokkaido 080‑8555, Japan

Tel.: +81-155-49-5648

Reviewer 2 Report

The manuscript reports on genes expressed in longhorn ticks infected with Babesia microti protozoans, a human pathogen. While it is clear that the authors have publishable data, they seem to have rushed a manuscript into submission. The manuscript does not have a materials and methods section and on that omission alone it is not publishable. There are some other serious flaws that need to be addressed when the authors resubmit. 

Because the manuscript was hastily thrown together it lacks clarity on what the authors did and what they found. And because there is no materials and methods as a reviewer I am unable to suggest corrections in the abstract, the introduction and the results that would help clarify the reporting.

In the abstract, for example, the authors need to clairfy if the ticks were adults nymphs or larvae. This difference is crucial in the transmission of apicomplexan protozoans. The authors confound fed and infected ticks with unfed and non-infected ticks in the abtract. Again, these differences are crucial for understanding transmission and the upregulation of genes. 

In the introduction there are nonsensical statements that "mites confer resistance to types of acaricides." No they don't, and why are we writing about mites? Apparently the authors were too rushed to learn about and cite articles on acaricide resistance in ticks, which is a global problem with a vast literature. 

The figures are unacceptable. Figs. 1 and 3 are beautiful and clear. But the information they portray is trivial. Especially with the brief and inadequate legends. In contrast, Figs 2 an 4 convey really meaningful data, but are so reduced that they are unreadable.  Similarly, Table 2 provides no relevant information. It is an arbitrary classification of the function of the genes detected. 

Table 3 has data that compares differential gene expression and statistical significance of the differences in means. But looking at the means and standard deviations, the variance does not appear to be normal. This could be a sample size problem, or it could be the result of pseudoreplication. But N is not reported and again, without a methods section, a reader cannot evaluate this data.  

Author Response

(The authors gave the same response as above.)

Reviewer 3 Report

General Comments

This is an interesting work regarding the identification of H. longicornis genes expression in response to Babesia microti infection. Even though the data are quite novel, details on the methodology are lacking and should be addressed in order to make the paper more scientifically sound.

Specific Comments

  • What do the authors mean by “bissexual” ticks?
  • Lines 48-55: Were the authors talking about ticks or mites? Please, be consistent.
  • Lines 62 and 67: replace Babesia by B. microti. The results found in this study cannot be generalized by every Babesia species.
  • What are the main hosts for the immature and mature stages of H. longicornis?
  • What was the inoculum used for infection hamsters? I mean, hamsters were infected with rodents’ erytrocytes containing B. microti? In which parasitemia? What was the size of the inoculum?
  • How long did it take to reach 5% parasitemia?
  • How old were the female unfed ticks used in this experiment?
  • Line 279: twice
  • How long the female ticks fed on hamsters? Until drop-off?
  • Line 301: Part of the digested cDNA was ligated with Adapter 1 and part with the Adapter 2R. 
Please, be specific! How many microliters were used for each adapter?
  • Please, describe the ligation reaction.
  • How did the authors select the reference gene for qPCR assays? How many reference genes were used for selecting the best one? Which software was used for selection of the reference gene (GeNORM, NorMFinder)? Please, specify that. The selection of a reference gene without testing a bunch of others is not desirable. Each reference gene should be selected according to the specific experimental groups. Getting a reference gene from a previously published paper or previously conducted experiment is not suitable for this kind of study.
  • Please, specify the concentration of primers and reagents used in qPCR assays for each target gene. Did the authors perform gradient experiments in order to select the best primer and annealing temperature? How was conducted the Melting Curve in order to check the specificity of each qPCR assay? Please, check the MIQE guidelines from Bustin et al. (2009).
  • How about the efficiency of qPCR assays? Please, show the range of efficiency rates found in the qPCR assays.
  • What was the criteria used for chosing the delta Ct normalization method, according to the efficiency of the qPCR assays? I mean, did the authors achieve efficiencies around 100% in ALL qPCR assays?
  • How did the authors injected iRNA in the ticks?

Author Response

(The authors gave the same response as above.)

Round 2

Reviewer 1 Report

All my concerns were addressed in a positive way. As a result, the manuscript has improved. I recommend publication.

Reviewer 2 Report

The authors report important work on the molecular level interactions between asian longhorn ticks and Babesia parasites. They document the relationship between vitellogenin2 and calreticulin and infection with Babesia microti. 

the experimental design, the statistical analysis, the description of the methods and results are logical and well conceived. The text reflects a great expertise on the part of the authors. The revised version is much clearer than the first version in terms of presentation of the methods and results. 

There are only minor grammatical problems at lines 61-67 which could have better sentence structure. 

Reviewer 3 Report

Authors have addressed the main points raised by the reviewer.